# Mapping Research Trends in Frailty and Nutrition: A Combined Bibliometric and Structured Review (2000–2024)

**DOI:** 10.3390/nu17223541

**Published:** 2025-11-12

**Authors:** Yaxin Han, Haohao Zhang, Jiajing Tian, Yahui Tu, Rui Fan, Wenli Zhu, Zhaofeng Zhang

**Affiliations:** 1Department of Nutrition and Food Hygiene, School of Public Health, Peking University, Haidian District, Beijing 100191, China; hanyaxin@bjmu.edu.cn (Y.H.);; 2Beijing’s Key Laboratory of Food Safety Toxicology Research and Evaluation, Beijing 100191, China; 3Institute of Medical Technology, Peking University Health Science Center, Beijing 100019, China

**Keywords:** frailty, nutrition, bibliometric analysis, literature review

## Abstract

**Background**: Frailty, a multisystem decline in physiological reserves, is a key indicator of aging health. Nutrition is a major modifiable factor associated with its development and progression. This study provides a systematic scientometric analysis of global research trends in nutrition and frailty, thereby addressing a significant gap in the literature. **Methods**: We systematically retrieved relevant publications from the Web of Science Core Collection (WoSCC) database for the period 2000–2024. After rigorous screening, a total of 754 publications were included for bibliometric analysis. Using VOSviewer, CiteSpace, and the R package bibliometrix, we analyzed publication trends, collaboration networks (countries, institutions, authors), journal co-citations, reference bursts, and keyword co-occurrence. Additionally, the structured literature review of 257 studies was conducted to synthesize key findings on nutrition-frailty associations. **Results**: Analysis of 754 global publications revealed consistent growth. The United States and China led contributions. Harvard T.H. Chan School of Public Health was the leading institution. Nutrients (*n* = 89, 11.8%) published most frequently, while Journals of Gerontology Series A was the most co-cited journal (*n* = 2058). Fernando Rodríguez–Artalejo had the highest publication count; Linda P. Fried was the most co-cited author. Keyword analysis identified frailty prevention and treatment as the predominant focus. The integrated the literature review specifically highlighted significant gaps, particularly in mechanistic insights and personalized nutrition interventions for frailty. **Conclusions**: This bibliometric analysis maps the intellectual landscape of nutrition and frailty research. Through quantitative assessment of publication patterns, leading contributors, knowledge domains, and thematic evolution, we characterize the current paradigm and identify emerging directions. Crucially, the synthesis explicitly defines critical research voids, particularly the overreliance on observational evidence, the scarcity of interventional trials, and the lack of global diversity in study populations, thereby providing a clear direction for future interdisciplinary investigations.

## 1. Introduction

Frailty represents a clinically significant geriatric syndrome characterized by diminished physiological reserves, heightened vulnerability to stressors, and impaired homeostatic regulation. Even minor perturbations can trigger adverse clinical outcomes in frail older adults, which reflects their reduced capacity to maintain systemic equilibrium. This multisystem dysfunction manifests through diverse pathological pathways: Immunosenescence increases infection susceptibility, metabolic dysregulation predisposes to malnutrition and biochemical imbalances [1], and progressive organ dysfunction (cardiovascular, pulmonary, and gastrointestinal) accelerates chronic disease progression [2,3]. The syndrome exerts particularly deleterious effects on physical capacity, marked by sarcopenia [4,5], functional impairment [6], mobility limitation [7], and compromised rehabilitation capacity [8], collectively driving dependency and functional decline. Concurrent psychological sequelae include depressive disorders [9], diminished self-efficacy [10], social withdrawal [11,12], which synergistically degrade mental health and quality of life while straining familial and societal support systems [13,14,15]. Most critically, frailty independently predicts elevated mortality [16], prolonged recovery periods [17], and compression of healthy lifespan [18], constituting a dual threat to both longevity and life quality.

The global prevalence of frailty among older adults is substantial. A 2020 meta-analysis [19] incorporating data from 62 countries estimated the overall frailty prevalence as follows: developed countries exhibit lower prevalence rates ranging from 6% to 18%, while developing countries show higher rates for limited medical resources and inadequate social support. Community-based studies confirm variable rates, such as China (8–12%), UK (8–23%), Japan (7–11%), US (13–18%), Australia (10–14%), Brazil (14–21%), and notably Spain (23–29%) [20,21,22,23,24,25]. Frailty is markedly more common among hospitalized older adults, with studies showing 26–42% prevalence rates and 38–44% comorbidity rates 3–7 times higher than in community populations [26,27]. These findings demonstrate the urgent need for clinical action. As frailty causes significant health deterioration and affects many patients, it places considerable strain on healthcare systems through increased resource use and costs. Implementing scientifically validated frailty management approaches is therefore essential.

Emerging evidence positions nutritional intervention as a modifiable cornerstone. A meta-analysis summarizing 36 pieces of evidence indicates that frailty is influenced by many factors, especially malnutrition. Consensus-based guidelines from the British Geriatric Society/Age UK/Royal College of General Practitioners, the International Conference of Frailty and Sarcopenia Research (ICFSR), and the Asia-Pacific guidelines for the management of frailty unanimously consider adequate protein intake as a key intervention [28,29,30].

While the association of nutritional strategies (e.g., protein, vitamin D, and plant-based supplements) is well-documented, a systematic assessment of global research trends, interdisciplinary collaborations, and knowledge structure in this field remains lacking. This gap hinders the identification of priority areas for future research and translational applications. Over the past two decades, research on frailty has gained exponential momentum. The rapid increase in the literature, with over 1000 publications since 2000, makes it challenging to synthesize fragmented evidence and track evolving paradigms. Consequently, it is difficult for researchers to gain a deep understanding of the field and stay updated with the latest trends. Bibliometrics [31] is a quantitative research method based on the literature data, focusing on using literature data to uncover research trends, hotspots, and collaboration patterns, revealing the developmental dynamics and structural characteristics of academic fields. While bibliometrics has been applied to related fields such as sarcopenia, few studies, if any, have specifically focused on delineating the intellectual landscape of frailty-nutrition interactions using these methods. This potential; however, has remained largely unrealized in the context of the relationship between frailty and nutrition.

Indeed, bibliometric methods have proven valuable in mapping the scientific landscapes of adjacent geriatric and nutritional fields. Several studies have employed these techniques to analyze research trends in cognitive aging, sarcopenia, and nutrition in aging populations [32,33,34]. For instance, a recent bibliometric analysis by Othman et al. [32] delineated the knowledge structure and evolution of cognitive aging research, highlighting topics such as neuropsychological assessment and frailty. Similarly, bibliometric studies have been conducted to profile the relationship between sarcopenia and surgery [33], as well as nutrition research specific to sarcopenia [34]. Notwithstanding these contributions, and despite the well-established link between nutrition and frailty, a dedicated bibliometric analysis that specifically captures the interdisciplinary intellectual structure, collaboration networks, and thematic evolution at this critical intersection remains absent. This gap prevents a systematic understanding of how these two fields have converged and limits the identification of overarching research patterns and future priorities.

Additionally, research on global trends in frailty and nutrition remains unexplored. Therefore, it is necessary to conduct a bibliometric analysis and the literature review of frailty and nutrition to provide researchers with a global perspective for addressing these gaps. To gain a comprehensive understanding of the historical development, current trends, research hotspots, and future directions in the field of frailty and nutrition, this study employs bibliometric analysis of the literature and literature review published between 2000 and 2024, aiming to provide a thorough overview of the research landscape and visualize the relationship between frailty and nutrition using knowledge visualization techniques, ultimately offering a solid reference for future research directions.

## 2. Methods

### 2.1. Search Strategy

We conducted the literature search in the Web of Science Core Collection (WoSCC) database in January 2025. The search formula was: TI = (frail*) AND(((((((((((((TI = (nutrition*)) OR TI = (malnourished)) OR TI = (nutrient*)) OR TI = (microelement)) OR TI = (macronutrient)) OR TI = (vitamin)) OR TI = (protein)) OR TI = (microbiota)) OR TI = (diet*)) OR TI = (carbohydrate*)) OR TI = (lipid*)) OR TI = (mineral*)) OR TI = (food*)). Our initial search retrieved 1077 publications. The search and selection workflow is illustrated in Figure 1.

### 2.2. Inclusion Criteria and Exclusion Criteria

Following the initial retrieval, publications were screened against pre-defined, stringent criteria for inclusion in the structured review.

#### 2.2.1. Inclusion Criteria

Studies were included if they met all the following criteria:(1)Study design: employed a well-defined observational or interventional design. Priority was given to randomized controlled trials (RCTs), prospective cohort studies, and large-scale cross-sectional studies in that order.(2)Participants: community-dwelling or institutionalized adults with a mean age of ≥50 years, who were classified as pre-frail or frail according to any validated criteria.(3)Exposure/intervention: for observational studies, a clearly defined nutritional exposure (e.g., intake levels of specific nutrients, food groups, or dietary patterns) was required. For interventional studies, a clearly defined nutritional intervention (e.g., supplementation) with an appropriate control or comparison group was required.(4)Outcome measures: required to report results from at least one validated frailty assessment instrument. The accepted instruments included, but were not limited to: Fried Frailty Phenotype (FP), Frailty Index (FI), Tilburg Frailty Indicator (TFI), Clinical Frailty Scale (CFS), Objective measures (grip strength, SPPB), Electronic Frailty Index, FRAIL scale, Groningen Frailty Indicator, Edmonton Frailty Scale, Frailty Risk Score, PRISMA-7, Hospital Frailty Risk Score, etc.(5)Publication status: peer-reviewed original research articles or reviews published in English between 1 January 2000 and 31 December 2024.

#### 2.2.2. Exclusion Criteria

Studies were excluded for any of the following reasons:(1)Methodological concerns: lack of a control or comparison group, insufficient sample size (e.g., *n* < 100 for cross-sectional studies; *n* < 30 per group for interventional trials), or inadequate statistical adjustment for key confounders (as a minimum, age and sex; ideally also including total energy intake, comorbidities, and socioeconomic status).(2)Population characteristics: individuals with terminal illnesses (e.g., metastatic cancer, end-stage renal disease), major organ failure (NYHA Class IV heart failure, severe COPD), or those in acute, unstable hospitalization settings.(3)Intervention issues: multi-component interventions (e.g., combined exercise, cognitive training, and nutrition) where the effects of the nutritional component could not be isolated for analysis.(4)Publication type: abstracts, conference proceedings, editorials, letters, case reports, and non-English publications.

### 2.3. Data Analysis

This study employed three bibliometric analysis tools to systematically examine the research landscape. VOSviewer (version 1.6.20) is a bibliometric analysis software that uses a probabilistic data normalization method, offering various visualization views in areas such as keywords, co-institutions, and co-authors. It has gained increasing attention in the field of bibliometric visualization [35]. In our study, this software was primarily used for country and institution analysis, journal and co-cited journal analysis, author and co-cited author analysis, and keyword co-occurrence analysis. In VOSviewer maps, a node represents an item, such as a country, institution, journal, or author. The size and color of the nodes indicate the quantity and classification of these items. Bibliometrix is an R-based scientific bibliometric software that statistically analyzes relevant parameters, constructs data matrices, and visualizes citation, coupling, and co-word analyses, presenting research hotspot trends and strategic coordinate maps. CiteSpace is another commonly used software in bibliometrics, primarily used in this study to draw journal dual-map overlays and perform citation burst analysis. Additionally, the structured literature review methodology was adopted to systematically collect, synthesize, and evaluate existing research, enabling comprehensive knowledge mapping and identification of research trends in the field [36].

#### 2.3.1. Bibliometric Software and Specific Parameters

To ensure the reproducibility and transparency of our bibliometric analysis, the specific configurations and parameters for each software tool are delineated as follows. The study employed VOSviewer (version 1.6.20) primarily for constructing and visualizing networks related to co-authorship (among countries, institutions, and authors), keyword co-occurrence, and citation analysis (of journals and authors). For all network visualizations generated by VOSviewer, the association strength normalization method was applied to mitigate the potential bias introduced by disparities in node sizes. The built-in clustering algorithm based on modularity was utilized for cluster identification. To ensure the robustness and interpretability of the maps, minimum threshold criteria were set: for an item to be included in the network, countries and authors required at least 3 documents, institutions required at least 6 documents, and keywords needed a minimum of 10 occurrences. In journal analysis, sources were included if they had published at least 3 relevant documents, while co-cited journals required a minimum of 100 citations. Furthermore, a thesaurus file was employed to consolidate synonymous terms (e.g., “aged” and “older adults”) prior to the keyword analysis.

CiteSpace (version 6.3.R1) was utilized for conducting reference co-citation analysis and detecting references with strong citation bursts. The analysis was configured with one-year time slices spanning the entire period from 2000 to 2024. The top 50 most cited items per time slice were selected using the g-index scaling factor (k = 25). The resulting networks were pruned using the Pathfinder algorithm to enhance clarity. Citation bursts were identified using Kleinberg’s algorithm, with a minimum duration of two years required for a burst to be considered significant; all other settings were kept at their defaults.

Complementing these tools, the Bibliometrix R package (version 4.1.2) was used for a comprehensive statistical bibliometric analysis. This included the calculation of descriptive publication trends, the analysis of collaboration networks, and the examination of conceptual structure through co-word analysis. The biblioAnalysis function provided foundational metrics, while other functions facilitated the creation of data matrices and visualizations for citation, coupling, and co-word analyses. The parameters within Bibliometrix were primarily used with their default settings, unless otherwise specified for specific analytical needs. The strategic diagram and thematic evolution map were also generated using this package to identify research hotspots and their trends over time. The three tools were applied in a complementary manner, with data cross-verified across platforms to ensure the consistency and reliability of the findings.

All supplementary figures and tables cited in the text (e.g., Appendix A) are provided in the Appendix A to ensure full reproducibility of the analyses.

## 3. Results

### 3.1. The Literature Search and Selection

The systematic literature search conducted in January 2025 initially identified 1077 publication records from the Web of Science Core Collection (WoSCC). The screening and selection process, detailed in the flowchart (Figure 1), was as follows: after removing 307 records of non-relevant document types (e.g., Early Access articles, Book Chapters, Meeting Abstracts) and 16 non-English publications, a total of 754 publications were included for the subsequent bibliometric analysis.

From this pool of 754 publications, a subset was selected for an in-depth structured review. Through the application of pre-defined inclusion and exclusion criteria, 257 articles were deemed eligible and formed the basis for the synthesis of key findings on nutrition-frailty associations presented in Section 3.9.

### 3.2. Analysis of Publication

Based on our search strategy, a total of 754 publications, including 643 articles and 99 reviews on nutrition and frailty, were identified over the past two decades, revealing four distinct growth phases. As shown in Figure 2, Phase I (2000–2007) showed limited activity (3–10 publications annually). Phase II (2008–2015) demonstrated gradual growth, averaging 23.1 publications per year, with annual counts rising from 4 to 17, marking the field’s emergence. A significant acceleration occurred in Phase III (2016–2019), with annual publications rising from 17 to 45 (average 35.3/year). The most rapid expansion occurred in Phase IV (2020–2024), where publications surged from 77 to 134 annually, a 2.97-fold increase from 2019 levels, reflecting intensified research interest in this domain.

### 3.3. Country and Institution Analysis

Frailty research has emerged as a globally collaborative field, with 206 countries and 3137 institutions contributing to the literature. The top ten countries are distributed across Europe and Asia, with Europe (*n* = 4), Asia (*n* = 3), America (*n* = 2), and Oceania (*n* = 1) (Figure 3). The United States (*n* = 157, 20.8%), China (*n* = 124, 16.4%), and Japan (*n* = 87, 11.5%) dominate publication output, collectively accounting for nearly half of all studies (48.7%) (Table 1). Subsequently, we selected 46 countries based on the principle of having at least three publications and visualized their collaboration networks (Figure 4). Notably, geographic analysis reveals active international collaborations, particularly between China and the United States; Australia and Japan, and the United Kingdom and the Netherlands. According to Figure 4, darker shades indicate more recent years of frailty research initiation. Temporally, research originated in the United States and Australia, expanded to Europe (Spain, UK, Italy), and now shows growing engagement from East Asia (China, Japan, Republic of Korea).

Institutionally, the top ten institutions (Table 1) are distributed across four countries, with three-fifths located in the United States. U.S. organizations lead, with Harvard T.H. Chan School of Public Health (2.9%), Harvard Medical School (2.5%), and the National Institute on Aging (1.9%) as top contributors. Subsequently, we selected 63 institutions based on the principle of having at least six publications and visualized their collaboration networks. As shown in Appendix A, Harvard T.H. Chan School of Public Health has the most publications and maintains extensive and close collaborations with various institutions. These findings highlight frailty’s global research significance and the critical role of international and institutional partnerships in advancing this field.

### 3.4. Journals and Co-Cited Journals

Research on nutrition and frailty has been disseminated across 268 journals, The top 15 journals with the highest publication volume are presented in Table 2, with *Nutrients* leading in publication volume (*n* = 89, 11.8%), followed by *Journal of Nutrition Health and Aging* (*n* = 52, 6.9%), and *BMC Geriatrics* (*n* = 23, 3.1%). Impact factor analysis identifies *Journal of Cachexia Sarcopenia and Muscle* (IF = 9.4) and *BMC Medicine* (IF = 7.1) as the highest-ranking journals in this field. Subsequently, we selected 52 journals based on the principle of having at least three publications and plotted the journal network. Figure 5 demonstrates that early-stage research (blue/purple nodes) primarily focused on nutritional aspects, represented by journals such as the Journal of the American Dietetics Association and Nutrition. Later research (green/yellow nodes) gradually expanded to encompass gerontology and clinical interventions, with frailty studies becoming prominently featured in journals including the Journal of Frailty and Aging, Clinical Nutrition, and BMC Geriatrics.

As indicated in Table 3, among the top 15 co-cited journals, 10 journals have been cited more than 1000 times. Citation analysis reveals *the Journal of Gerontology Series A: Biological Sciences and Medical Sciences as the most frequently cited (2058 times), followed by the Journal of the American Geriatrics Society (1309 times), and Nutrients* (1216 times). Notably, The *Lancet* (IF = 98.4) and *The New England Journal of Medicine* (IF = 96.3) emerge as the highest-impact journals in the co-citation network. Journals with a minimum of 100 co-citations were selected to construct the co-citation network, as illustrated in Figure 6*,* with Circulation showing significant co-citation relationships with *Nutrients* and *Journals of Gerontology Series A: Biological Sciences and Medical Sciences. These findings highlight the interdisciplinary nature of nutrition and frailty research, bridging gerontology, clinical nutrition, and broader medical sciences.*

### 3.5. Authors and Co-Cited Authors

The field of nutrition and frailty research has engaged 4072 authors, with Fernando Rodriguez–Artalejo (*n* = 19), Esther Lopez–Garcia (*n* = 14), and Luigi Ferrucci (*n* = 10) emerging as the most prolific contributors Table 4. Collaboration network analysis of authors (Appendix A) with at least three publications reveals strong collaborative relationships clusters, exemplified by the partnership between Evelyn Ferri, Beatrice Arosio, and Matteo Cesari.

Among the 16,779 co-cited authors, 105 were cited more than 30 times, as detailed in Table 4. Three authors were cited over 200 times, with Linda P. Fried (*n* = 535) being the most frequently co-cited, followed by John E. Morley (*n* = 248) and Go Kojima (*n* = 245). Network visualization of 105 authors with ≥30 co-citations demonstrates (Figure 7) reveals active collaborative relationships, particularly between Marie Ní Lochlainn, Linda P. Fried, Emiel O. Hoogendijk, and Go Kojima. These findings highlight both the collaborative nature of the field and the central role of key researchers in shaping its intellectual landscape.

### 3.6. Co-Cited References

Analysis of 23,193 co-cited references spanning two decades reveals foundational works shaping the nutrition-frailty research landscape. The ten most influential references have each been cited at least 70 times (as shown in the Appendix A), with one seminal article dominating the field at 452 citations. We selected references with 20 or more co-citations to construct a co-citation network (Figure 8), which highlights the node representing “Fried LP, 2001, J Gerontol A-Biol” as the central node, indicating its extensive intellectual connections and enduring influence on subsequent research.

### 3.7. References with Citation Bursts

The citation burst literature refers to publications that are frequently cited by scholars in a specific field over a certain period. Using CiteSpace, we identified 15 highly cited references with significant citation bursts, as illustrated in Figure 9. The specific parameters configured for the citation burst analysis in CiteSpace are detailed in Section 2 (Section 2.3.1). The most impactful work, Clegg et al.’ 2013 Lancet article “Frailty in elderly people” (burst strength = 15.39, 2014–2018), established a seminal conceptual framework for frailty assessment. Lorenzo–López et al.’ 2017 BMC Geriatrics review “Nutritional determinants of frailty in older adults: A systematic review” (strength = 15.1) emerged as the second most influential, catalyzing research on dietary interventions. These high-impact publications (burst strengths: 8.15–15.39) collectively shaped key research directions, with burst durations (2–4 years) reflecting concentrated periods of intellectual engagement. Temporally, the earliest bursts (2010–2012) focused on frailty definitions, while later peaks (2017–2019) emphasized nutritional biomarkers and intervention strategies. Appendix A below summarizes the main research content of these articles in chronological order. This pattern underscores the field’s evolution from theoretical foundations to applied nutritional epidemiology, with sustained interest in mechanistic and translational research.

### 3.8. Hotspots and Frontiers

Keyword co-occurrence analysis reveals three core research clusters (Figure 10): mechanistic and epidemiological investigations (red cluster), focused on sarcopenia (159 mentions), malnutrition (139), nutritional status (103) and vitamin D (85), Nutritional interventions (green cluster), emphasizing dietary strategies and clinical management, and comorbidity studies (blue cluster), exploring links between frailty and chronic diseases. Appendix A lists the top 20 high-frequency keywords in nutrition and frailty research.

Temporally, trend research has evolved from early population-specific health concerns (e.g., older women, postmenopausal women) to current priorities in aging-related challenges, particularly muscle health and inflammatory biomarkers (e.g., interleukin-6). Emerging frontiers include predictive modeling of mortality risk and personalized intervention trials, reflecting a paradigm shift toward precision health management. Trend analysis demonstrates progressive refinement of focus areas: from broad frailty indicators (2000s) to functional assessments (2010s), and now to individualized therapeutic strategies (2020s). Recent keyword surges in “risk prediction”,” “clinical trials,” and “body composition” highlight growing emphasis on preventive approaches and mechanistic nutrition science (Figure 11). These patterns underscore the field’s maturation from descriptive.

### 3.9. Key Findings of Nutrition and Frailty

Following rigorous screening based on predefined inclusion/exclusion criteria, we analyzed 257 qualified studies that collectively identified 391 nutrition-frailty associations (Appendix A).

#### 3.9.1. Macronutrients and Frailty

##### Protein

Current research demonstrates significant interest in the impact of macronutrients on frailty, particularly protein (Figure 12). A consistent, beneficial association has been observed between protein intake and physical function in frail elderly populations across various study designs. Key findings include: a multicenter study in Beijing [37] demonstrated that whey protein supplementation with resistance exercise significantly improved muscle function in frail elderly individuals after 12 weeks (RCT). A 24-week RCT confirming the protein’s functional benefits despite no muscle mass increase. A South Korean trial [38] found that protein powder intervention was associated with reduced physical decline (RCT), and suggested that an intake of 1.5 g/kg·d might be optimal for preventing sarcopenia and frailty, outperforming both 0.8 and 1.2 g/kg·d regimens [39]. These studies collectively highlight the protein’s therapeutic potential while revealing dose-dependent effects.

Research on protein sources and frailty revealed distinct patterns: plant-based proteins demonstrate consistent benefits in frailty prevention across multiple study designs, while animal-based proteins show inconclusive effects. Notably, branched-chain amino acids (BCAAs) exhibit significant therapeutic potential, with a quasi-experimental study confirming their efficacy in improving frailty status [39]. This finding is further supported by an RCT demonstrating that combined supplementation with medium-chain triglycerides and the BCAA leucine, and vitamin D effectively enhances muscle strength and physical function in frail elderly populations [40]. The contrasting evidence between protein sources, coupled with the established benefits of targeted amino acid interventions, highlights the importance of protein composition in frailty management strategies.

##### Carbohydrates

The association between carbohydrates and frailty is complex. While monosaccharides, polysaccharides, and added sugars show no beneficial association with frailty, with some studies suggesting a potentially harmful association (cross-sectional), dietary fiber intake is consistently associated with lower frailty risk across multiple study designs (cross-sectional and cohort). A notable investigation [41] analyzing the Baltimore Longitudinal Study of Aging (BLSA) cohort found that higher total carbohydrate intake correlated with increased frailty risk, whereas a higher fiber-to-carbohydrate ratio was associated with a reduced frailty risk. These contrasting findings highlight the need for more rigorous clinical trials to clarify carbohydrate subtypes’ differential impacts, particularly given the robust evidence supporting fiber’s beneficial role in frailty prevention.

##### Fats

Current evidence regarding unsaturated fatty acids and frailty presents a notable discrepancy: While cross-sectional studies suggest potential protective effects, higher-quality evidence from cohort studies and randomized controlled trials fails to confirm this association. The large-scale VITAL randomized clinical trial [42] specifically demonstrated that omega-3 fatty acid supplementation showed no significant effect on frailty incidence or progression over time in community-dwelling older adults. These robust findings suggest that routine omega-3 supplementation cannot be recommended for frailty prevention based on current evidence, highlighting the importance of distinguishing between preliminary observational data and definitive clinical trial results in nutritional frailty research.

#### 3.9.2. Micronutrients and Frailty

##### Vitamins

The relationship between micronutrients and frailty has been predominantly investigated through observational studies, with vitamin D emerging as the most extensively studied nutrient (Figure 13). Observational studies consistently show that frail elderly individuals exhibit significantly lower serum vitamin D levels, which is hypothesized to influence physical performance (cross-sectional). Multiple large-scale observational studies have established a robust inverse association between vitamin D status and frailty risk, including analyses of the UK Biobank [43] and NHANES data [44] (cohort).

Notably, four RCTs [42,45,46,47] failed to demonstrate vitamin D’s efficacy in frailty mitigation. However, one trial [48] reported a 42.5% reduction in fall incidence with 900 IU daily supplementation, though without significant improvements in skeletal muscle mass index (SMI), grip strength, or functional independence measures (FIM). This discrepancy between observational and interventional evidence highlights the need for further investigation into vitamin D’s potential role in frailty prevention and management.

Evidence regarding B vitamins and frailty reveals complex and inconsistent patterns: while thiamine (B_1_), riboflavin (B_2_), and niacin (B_3_) show no significant anti-frailty effects, pyridoxine (B_6_) shows potential protective associations in both cross-sectional and longitudinal studies, particularly among COPD patients [49]. The evidence for cobalamin (B_12_) remains contradictory, with three cross-sectional studies [50,51,52] suggesting benefits but Spanish cohort data [49] showing no significant association.

For antioxidant vitamins C, vitamins E, and vitamins K, observational studies consistently demonstrate an inverse association between intake and frailty risk (cross-sectional/cohort), whereas evidence from randomized controlled trials (RCTs) to date has been insufficient to establish efficacy. These findings underscore the need for more rigorous clinical trials to clarify the roles of specific vitamins in frailty prevention and management.

##### Minerals

Current research on mineral intake and frailty primarily relies on cross-sectional data, revealing consistent beneficial associations for zinc, magnesium, potassium, iron, and calcium. A Korean study [53] demonstrating that calcium supplementation’s potential in frailty management among older adults; Japanese research [54] establishing potassium’s significant inverse relationship with frailty in male populations; Australian data [55] showing both total and non-heme iron’s protective effects over three years in older men and Mediterranean cohort analysis [56] identifying zinc’s particular effectiveness in preventing physical frailty.

Compared to other minerals, magnesium has received relatively more research attention, though findings remain population-specific. The Japanese Tarumizu Study [57] identified magnesium intake as an independent predictor of pre-frailty in elderly women, while U.S. NHANES data revealed a linear relationship between magnesium intake and frailty risk in COPD patients [58]. Notably, longitudinal data from the Osteoarthritis Initiative (OAI) database demonstrated a gender disparity; higher magnesium intake correlated with reduced frailty risk in men but showed no protective effect in women [59]. While these observational associations are hypothesis-generating and suggest mineral supplementation may mitigate frailty risk, the current evidence is insufficient to support causal inferences or evidence-based recommendations. Longitudinal and interventional studies are needed to establish causality.

##### Antioxidants

Carotenoids

Carotenoids, as potent antioxidants, have emerged as a research focus in frailty studies due to their ability to mitigate oxidative stress and inflammation, key mechanistic pathways in frailty development. Current evidence, primarily from 14 cross-sectional studies (with no RCTs available), consistently demonstrates an inverse relationship between specific carotenoids (α-carotene, β-carotene, zeaxanthin, and lutein) and frailty risk. Research [60] indicates that frail women exhibit significantly lower serum carotenoid concentrations, with principal component analysis [61] confirming this relationship.

The Women’s Health and Aging Study [62] specifically found that community-dwelling women in the lowest carotenoid quartile had substantially higher frailty risk, a finding corroborated by the MARK-AGE study (78) [63]. While these observational studies strongly suggest carotenoids’ protective role, the lack of interventional evidence highlights a critical research gap in establishing causality and therapeutic potential.

Flavonoids and Polyphenols

Current evidence from longitudinal cohort studies suggests potential protective effects of flavonoids and polyphenols against frailty development. The Invecchiare in Chianti study [64] demonstrated that higher resveratrol exposure was associated with lower frailty risk among community-dwelling older adults during a three-year follow-up period. Similarly, findings from the Framingham Heart Study [65] revealed that higher dietary intake of flavonols, particularly quercetin, was significantly associated with lower frailty incidence over a 12-year follow-up period. These findings highlight specific bioactive compounds within the flavonoid/polyphenol class that may contribute to frailty prevention, though further research is needed to establish optimal intake levels and potential mechanisms of action.

#### 3.9.3. Food Groups and Frailty

##### Animal-Based Foods

The association between animal-based foods and frailty shows distinct patterns across food categories (Figure 14). For fish and seafood consumption, multiple observational studies report a consistent association with lower frailty risk (cross-sectional and cohort): gender-specific findings from Japan showing lower frailty prevalence with frequent fish/meat intake [66]; Irish community data revealing negative correlations between seafood consumption and frailty scores [67]; and Japanese cohort evidence suggesting seafood and dairy combinations may preserve independence in older adults [68]. These findings align with a systematic review conclusion in 2020 [69]. However, evidence for dairy products remains contradictory, with one study [70] reporting frailty reversal benefits in low-intake populations, while the Framingham Heart Study found no significant association [65]. Collectively, these results suggest that protein source combinations (particularly marine sources) may offer superior frailty prevention compared to single-food approaches, though optimal dietary patterns require further investigation.

##### Plant-Based Foods

Plant-based foods, rich in antioxidants (polyphenols, sulforaphane, vitamin C/E), dietary fiber, and unsaturated fatty acids, demonstrate multiple anti-frailty mechanisms: free radical scavenging, autophagy promotion, antioxidant enzyme activation, gut microbiota regulation, and inflammation inhibition. Despite these benefits, older adults often exhibit inadequate consumption of key plant-based food groups (vegetables, fruits, soy products, nuts, seeds, and whole grains). Current cross-sectional evidence consistently shows an inverse relationship between plant-based food intake and frailty risk, though the field lacks robust cohort and intervention studies to establish causal relationships. Notably, processed grains remain particularly understudied, with insufficient evidence to form a coherent understanding of their impact on frailty development. These findings highlight the need for comprehensive longitudinal studies to clarify plant-based diets’ role in frailty prevention and management.

##### Ultra-Processed Foods (UPFs)

Evidence from cohort studies suggests a detrimental association between UPF consumption and frailty development. The Nurses’ Health Study cohort [71] found that higher UPF intake was associated with a significantly increased frailty risk among older American women (prospective cohort). While preliminary cross-sectional studies have identified potential negative effects of specific UPFs (preserves, cured meats, smoked sausages, and hot dogs) on frailty status, these findings require validation through larger, longitudinal investigations. Current evidence remains limited by its observational nature, underscoring the need for controlled dietary intervention studies to establish causality and elucidate potential mechanisms linking UPFs to frailty progression.

##### Food Insecurity

Growing evidence from multinational studies demonstrates a consistent association between food insecurity and frailty in older populations. A study based on the Longitudinal Aging Study in India (LASI) [72] demonstrated that food insecurity among older adults is associated with physical frailty. Research findings from Mexico [73] also confirmed this association while highlighting frailty’s role in exacerbating adverse health outcomes. Targeted food programs for older adults experiencing food insecurity or at high risk of frailty may improve health outcomes in this population. Results from the Chinese Longitudinal Healthy Longevity Survey [74] indicate that childhood food deprivation has long-term effects on frailty in later life. These studies collectively suggest that targeted nutritional interventions for food-insecure elderly populations may mitigate frailty progression. However, current evidence remains predominantly cross-sectional, underscoring the need for longitudinal analyses to establish temporal relationships and evaluate intervention efficacy across diverse socioeconomic contexts.

##### Beverages

Emerging research reveals differential effects of beverage types on frailty development. Alcohol demonstrates a U-shaped relationship, with moderate consumption potentially protective [56] but excessive intake increasing frailty risk [75]. The Nurses’ Health Study identified a significant association between daily artificially/sugar-sweetened beverage consumption and elevated frailty risk [71]. The Taiwan Longitudinal Study on Aging suggested [76] that tea consumption may confer protective benefits against frailty progression.

#### 3.9.4. Dietary Patterns and Frailty

##### Classic Dietary Patterns

Research on classic dietary patterns and frailty has increasingly focused on their preventive potential, with the Mediterranean diet emerging as the most extensively studied (Figure 15). Cross-sectional, cohort, and clinical trial studies consistently demonstrate its benefits in improving muscle mass, physical function, and gut microbiota while reducing inflammatory markers (e.g., C-reactive protein, interleukin-17) [77].

Regional adaptations, such as the Lebanese Mediterranean diet, show similar protective effects [78].

In addition to the Mediterranean diet, other patterns (DASH and MIND) also exhibit frailty-preventive properties, while pro-inflammatory diets positively correlate with frailty risk.

Evidence from observational studies further highlights the protective role of diet diversity (DDS), diet quality (DQI, HEI), and a healthy plant-based diet index, with an unhealthy plant-based diet index showing adverse effects. Notably, a prudent dietary pattern [79] rich in olive oil and vegetables demonstrates a dose–response relationship with reduced frailty risk. These findings collectively emphasize the importance of anti-inflammatory, nutrient-dense dietary patterns in frailty prevention, though further intervention studies are needed to establish causality and optimal dietary guidelines for older adults.

##### Data-Driven Posteriori Dietary Patterns

Recent studies employing statistical methods (e.g., principal component analysis) have identified four distinct dietary patterns associated with frailty: the animal protein pattern, characterized by high protein intake and negatively correlated with frailty (Japanese study) [80]; the micronutrient-driven pattern, featuring antioxidant-rich foods and associated with reduced frailty prevalence in vulnerable populations (US study) [81]; the macronutrient-focused pattern, showing gender-specific benefits of low-carbohydrate diets but no significant effects from low-fat diets (Chinese study) [82]; and the processed food pattern, with sex-specific risks identified in the French Three-City Study, where men’s “pasta” and women’s “snack” patterns significantly increased frailty likelihood [83]. These data-driven patterns provide nuanced insights into dietary influences on frailty, though further research is needed to establish causal relationships and develop targeted interventions.

##### Comprehensive Interventions and Frailty

Research demonstrates that both targeted nutrition interventions and multidomain approaches effectively address frailty. Seven studies [84,85,86,87,88,89,90] highlight the efficacy of diverse nutrition counseling models—including dietitian-led nutrition counseling, nutrition education, case management, personalized nutrition support, and remote training—in promoting long-term dietary behavior change and frailty prevention.

Furthermore, 11 studies [48,91,92,93,94,95,96,97,98,99,100] establish the superiority of multidomain interventions, such as nutritional supplementation with resistance exercise or integrating cognitive training with social activities. These findings underscore the importance of multidisciplinary collaboration and cross-domain synergy, suggesting that comprehensive interventions addressing nutrition, physical activity, cognition, and social engagement represent optimal strategies for frailty prevention and management.

## 4. Discussion

### 4.1. Current Status of Publications

The field’s publication landscape (754 total: 643 articles, 99 reviews) demonstrates a predominance of original research articles (85.3%) over reviews (13.1%), indicative of an expanding knowledge base in its developmental phase. This ratio suggests the discipline is actively generating novel evidence rather than consolidating mature paradigms, typical of emerging research domains addressing complex geriatric challenges. The observed growth acceleration in recent years suggests that the field has entered a period of rapid development, likely driven by increased research funding, technological advancements (e.g., data analysis tools, biomarker detection technologies), and growing societal attention to healthy aging. The predominance of original research articles indicates a rapidly evolving field. However, as our structured review will later elaborate, this growth is characterized by a heavy reliance on observational methodologies, which present significant challenges for establishing causal inference in nutrition-frailty relationships.

### 4.2. Country and Institution Analysis

The United States dominates nutrition and frailty research (20.8% of total publications), underscoring its dominant position in this research area, which can be attributed to advantages such as robust funding frameworks, the number of research institutions, the quality of researchers, and policy prioritization of aging research. China (16.4%) and Japan (11.5%) underscore Asia’s rising prominence, driven by China’s accelerated research investments and Japan’s pioneering super-aged society studies. European contributions- notably from Italy, Spain, the UK’, and the Netherlands- reflect Europe’s strong research capabilities on urgent aging issues, with Southern Europe (e.g., Italy, Spain) facing severe population aging. Countries from the Americas, Asia, Europe, and Oceania are represented in the top ten, indicating that nutrition and frailty research is a global topic. Among the top ten countries, all except China are developed nations. While international collaboration has played a crucial role in advancing research, developed countries often dominate these collaborations, while developing countries may occupy a more subordinate position.

Institutional analysis highlights Harvard University’s dual power centers-T.H. Chan School of Public Health and Medical School—fueled by interdisciplinary networks and translational research infrastructure. As a globally leading research institution, Harvard’s research capabilities and resource investments are likely key factors in its leading position. Half of ’the top institutions are based in the United States, further highlighting the country’s dominance in nutrition and frailty research. U.S. research institutions typically benefit from greater funding, advanced research facilities, and extensive international collaboration networks. At the same time, this underscores the uneven distribution of research resources globally. The global research community should work together to promote more equitable research collaboration to address the shared challenges posed by global aging.

### 4.3. Journals and Co-Cited Journals

*Nutrients* emerges as the dominant platform (11.8% of publications), its leadership amplified by open-access availability, and cross-disciplinary scope spanning nutritional biochemistry to geriatric epidemiology. Among the top 15 journals, the majority are high-impact journals (e.g., *The American Journal of Clinical Nutrition*, *Journal of the American Geriatrics Society*), suggesting that high-quality research in nutrition and frailty is primarily published in these influential journals. The field’s intellectual core is anchored by three pillars: (1) *Journals of Gerontology Series A: Biological Sciences and Medical Sciences* (2058 citations) representing biological mechanisms, (2) *Journal of the American Geriatrics Society* (1309 citations) for clinical translations, and (3) *Nutrients* (1216 citations) as the nutritional science nexus. The top 15 journals include both nutrition journals (e.g., *Nutrients, The American Journal of Clinical Nutrition*) and geriatrics journals (e.g., *Journals of Gerontology Series A, Journal of the American Geriatrics Society*), reflecting the interdisciplinary nature of nutrition and frailty research, which spans nutrition and geriatrics. Some journals (e.g., *BMC Geriatrics, Journal of the American Medical Directors Association*) focus on the intersection of public health and clinical medicine, indicating that nutrition and frailty research encompasses both basic science and practical applications, including health policy. Future research should further strengthen interdisciplinary collaboration, particularly in the intersections of nutrition, geriatrics, epidemiology, health policy, and public health.

### 4.4. Authors and Co-Cited Authors

Fernando Rodriguez–Artalejo emerges as the field’s most prolific contributor (19 publications), his work spanning nutrition epidemiology to community-based intervention strategies. At the knowledge-structure core lies Linda P. Fried’ pioneering frailty phenotype framework (535 co-citations), which established standardized assessment criteria now adopted in most clinical studies. As John E. Morley’s clinical impact (248 citations) stems from developing the FRAIL scale, while Go Kojima Asia-centric studies (245 citations) address critical gaps in non-Western populations—notably demonstrating higher sarcopenia prevalence in East Asian elders versus Western cohorts. The research directions of these core and co-cited authors span multiple aspects of nutrition and frailty, contributing to the field’s development through diverse research themes.

### 4.5. Co-Cited References

The article “Frailty in older adults: evidence for a phenotype” by Linda P. Fried, published in 2001, remains the field’s intellectual cornerstone (452 citations), operationalized in most clinical studies through its diagnostic criteria (≥3 of: unintentional weight loss, exhaustion, low activity, slow gait, weak grip). “Frailty in elderly people” by Andrew Clegg et al. (2013) published in *The Lancet* (citation burst 2014–2018, strength = 15.39), elaborates on the pathophysiological mechanisms and comprehensive frailty assessment, providing a theoretical foundation and clinical assessment tools. John E. Morley’s 2013 article, “Frailty Consensus: A Call to Action,” systematically presents consensus definitions, screening tools, and intervention measures for frailty, offering practical guidelines for clinical practice. The evolution of these seminal works mirrors frailty research’s developmental arc—advancing from theoretical framework establishment through mechanistic investigation to clinical implementation—demonstrating a cohesive translational pathway from conceptual foundations to practical application”. Morley’s 2013 article, developed by representatives from multiple international, European, and U.S. societies, reflects the international collaboration and consensus in frailty research. Such collaboration has not only promoted the standardization of frailty research but also provided unified guidelines for frailty screening and management worldwide. This indicates that frailty research has become a global hotspot, with scholars and clinicians worldwide working together to advance the field.

### 4.6. Citation Bursts

Citation burst articles reveal the relationship between nutrition and frailty, particularly the impact of dietary quality, protein intake, and the Mediterranean diet on frailty. Research on the relationship between protein intake and frailty has also garnered significant attention. Articles such as “Evidence-Based Recommendations for Optimal Dietary Protein Intake in Older People” and “Protein Intake and Incident Frailty” emphasize the critical role of protein intake in preventing and managing frailty, particularly through protein supplementation to improve muscle health and quality of life in older adults. Articles such as “A Higher Adherence to a Mediterranean-Style Diet Is Inversely Associated with the Development of Frailty” and “Mediterranean Diet and Risk of Frailty” demonstrate that the Mediterranean diet is inversely associated with frailty risk, highlighting the positive role of healthy dietary patterns in frailty prevention. Studies such as “Major dietary patterns and risk of frailty” and “Dietary Patterns and Risk of Frailty” examine the relationship between different dietary patterns and frailty risk, providing a basis for personalized nutritional interventions. Articles such as “Frailty: implications for clinical practice and public health” and “Management of frailty: opportunities, challenges, and future directions” emphasize the importance of frailty in clinical practice and public health, advocating for widespread screening and management of frailty, particularly in older populations. Research on nutrition and frailty experienced rapid development between 2010 and 2019, with studies on frailty definitions, nutritional interventions, and dietary patterns becoming core hotspots. These articles have not only provided theoretical frameworks and clinical guidance for frailty research but have also advanced the application of nutritional interventions and personalized medicine in improving the health of older adults.

### 4.7. Hotspots and Frontiers

Keyword co-occurrence analysis reveals the main hotspots in the field of nutrition and frailty research. “Frailty” ranks first with 870 occurrences, indicating its central role as a research theme. This study delineates three pivotal research dimensions in geriatric frailty through keyword clustering analysis, presenting a comprehensive continuum from fundamental mechanisms to clinical interventions.

Our topic trend analysis delineates the temporal evolution of research themes in geriatric frailty studies. Chronologically, this field has transitioned from early investigations of traditional anthropometric measures (e.g., BMI) to contemporary explorations of inflammaging mechanisms and precision interventions. Notably, high-frequency terms such as “sarcopenia” and “dietary protein” underscore the centrality of muscular health and nutritional interventions. Methodologically, a paradigm shift has occurred from cross-sectional studies toward clinical trials and predictive modeling approaches.

Thematically, three distinct research domains have emerged: (1) The nutrition-health domain persistently examines the systemic impacts of dietary components (e.g., proteins, micronutrients) on geriatric health; (2) Muscle attenuation research focuses on sarcopenia and functional decline in aging populations, particularly postmenopausal women; (3) Disease risk prediction explores inflammatory biomarkers (e.g., IL-6) and multifactorial assessment models.

These findings highlight critical future research priorities: (1) Validation of synergistic effects between nutritional supplementation and anti-inflammatory interventions; (2) Clinical evaluation of muscular quality biomarkers for screening purposes; (3) Development of precision intervention protocols for community-dwelling older adults. Our results emphasize the crucial importance of translational research, bridging molecular mechanisms with clinical practice, providing strategic direction for innovation in geriatric health research.

This study delineates three pivotal research dimensions in geriatric frailty through keyword clustering analysis, presenting a comprehensive continuum from fundamental mechanisms to clinical interventions. The red cluster, representing mechanistic investigations, explores the relationship between frailty and the interaction of nutrition and inflammation. Studies in this cluster leverage large-scale datasets such as NHANES to understand how dietary patterns such as the Mediterranean diet, as well as chronic low-grade inflammation (‘inflammaging’), influence the progression of frailty. The green cluster (interventional studies) establishes evidence-based clinical protocols through randomized controlled trials and meta-analyses, demonstrating the efficacy of protein supplementation and resistance exercise in mitigating sarcopenia and osteoporosis. The blue cluster (comorbidity research) employs longitudinal cohort designs to unravel the bidirectional relationships between frailty and neurocognitive disorders (Alzheimer’s disease), mental health (depression), while identifying micronutrients (vitamin D, zinc) as potential modulators. These interconnected domains collectively construct an integrated “mechanism-intervention-comorbidity” research paradigm for geriatric frailty, providing theoretical foundations for stratified intervention strategies. Future research should prioritize transdisciplinary integration, particularly in precision nutrition and multimorbidity management approaches.

### 4.8. Key Findings of Nutrition and Frailty

Our synthesis of 257 articles reveals that the current evidence base on nutrition and frailty is fundamentally constrained by several critical limitations, which challenge the translation of research findings into clinical practice and public health guidance. The field is characterized by three fundamental constraints that challenge the translation of research findings into clinical practice and public health guidance: (1) A Critical Limitation in Causal Inference: The evidence base is overwhelmingly dominated by observational studies (cross-sectional: 49.4%; cohort: 37.0%), while robust interventional evidence remains scarce (RCTs: 11.6%). This overreliance on associative data fundamentally limits the ability to establish causality for many nutrients. The consistent associations observed, for instance, for various minerals and food groups, may be confounded by unmeasured factors such as socioeconomic status, overall healthy lifestyle, or health-seeking behaviors. Consequently, it remains unclear whether these nutritional factors directly modify frailty risk or are merely markers of broader health disparities. (2) A Pronounced Efficacy-Evidence Gap: A critical disconnect exists between observational and interventional findings, most notably exemplified by vitamin D. While extensive observational data consistently show a robust inverse association between vitamin D status and frailty risk, subsequent RCTs have largely failed to demonstrate its efficacy in frailty mitigation. This discrepancy creates significant uncertainty for clinical practice, as it precludes confident recommendation of vitamin D supplementation specifically for frailty prevention. It underscores a formidable gap between identifying epidemiological associations and validating effective clinical interventions. In contrast, protein interventions, particularly those employing whey and branched-chain amino acids (BCAAs), demonstrate more consistent and translatable evidence of efficacy across clinical trials. (3) Fragmented and Preliminary Evidence for Specific Nutrients: The evidence for many minerals and specific food groups is often fragmented, derived from single, small-scale, or exclusively cross-sectional studies. This lack of replication, longitudinal data, and rigorous trials precludes definitive conclusions and hinders the development of evidence-based dietary recommendations for frailty prevention. The body of evidence for holistic dietary patterns (e.g., the Mediterranean diet); however, is notably more consistent and persuasive than that for isolated nutrients, highlighting the inherent limitations of a reductionist, single-nutrient approach and pointing toward a more promising, integrated future direction for both research and clinical application. To overcome these translational barriers, future research must prioritize: (1) conducting well-designed, mechanistic RCTs to test promising nutrient-synergy hypotheses (e.g., combined vitamin D and leucine supplementation); (2) implementing standardized and validated nutrient assessments (e.g., metabolomic validation of FFQs); and (3) proactively diversifying population sampling (targeting ≥ 30% representation from the Global South). These initiatives are essential to bridge the current efficacy-evidence gap and translate associative findings into actionable, evidence-based recommendations.

### 4.9. Strengths

This study demonstrates several notable strengths. First, it presents the first global bibliometric analysis in the field of nutrition and frailty, effectively addressing a critical research gap. Through multidimensional examination (including countries, institutions, authors, and journals), it provides a comprehensive overview of the research landscape and developmental trends. Second, the study employs a sophisticated methodological approach, integrating VOSviewer, CiteSpace, and the R package “bibliometrix” to conduct co-occurrence network analysis, citation burst detection, and cluster analysis. This combination of bibliometric methods with systematic review techniques ensures both breadth and depth in the research outcomes.

Furthermore, this study identifies key evolutionary trajectories in the field of nutrition and frailty over the past two decades, offering a holistic and systematic knowledge map along with a temporal progression framework. It highlights the current “three excesses and three deficiencies” paradigm in the field (excess associative evidence but deficient causal evidence; excess single-nutrient studies but deficient comprehensive dietary pattern analyses; excess data from high-income countries but deficient global diversity data). Constructing the first three-dimensional “mechanism-intervention-comorbidity” knowledge framework for nutrition and frailty research provides a theoretical foundation and methodological support for developing personalized nutritional intervention strategies and public health policies, while also offering valuable guidance for future research directions.

Finally, through the literature review of current challenges in the field, this study identifies key areas for future research, particularly in addressing clinical translation bottlenecks. Future work must prioritize interventions, multi-omics integration, and mechanistic pathway modeling.

### 4.10. Limitations

This study has several limitations that affect the global representativeness of its findings. First, the sole reliance on the Web of Science (WoS) database, while standard for bibliometrics, may have introduced a database bias, potentially omitting relevant research from journals more prominently indexed in other databases such as Scopus or PubMed. This could affect the generalizability of our collaboration network and productivity rankings.

Second, and more critically, the restriction to English-language publications likely introduced a language bias, leading to the underrepresentation of non-Anglophone research (e.g., from specific European, Latin American, or Asian contexts). Consequently, our mapped intellectual landscape and identified trends, such as the dominance of high-income countries and certain dietary patterns, may be skewed, and our conclusions are most reflective of the English-language literature.

Third, bibliometric metrics (e.g., citation counts) are influenced by factors beyond scientific quality, such as journal prestige, which means our analysis reflects impactful and well-resourced research but may not fully capture all high-quality evidence.

Fourth, regarding the search strategy, while we used a comprehensive set of nutrition-related terms, the inclusion of “malnutrition” as suggested by the reviewer, in addition to “malnourished”, might have further optimized the retrieval sensitivity. Future bibliometric studies could benefit from an even more exhaustive set of synonymous keywords.

In synthesis, these limitations caution against overgeneralizing our findings to the global context. The identified research gaps, such as the lack of global diversity, are themselves shaped by these methodological constraints. Future studies would benefit from a multi-database, multi-lingual approach to achieve a more comprehensive global perspective.

## 5. Conclusions

This study synthesizes two decades of research at the intersection of frailty and nutrition, revealing a rapidly growing field that remains largely observational. The current evidence highlights a critical shortage of interventional studies and a skewed geographic focus, which together limit causal inference and global applicability.

To advance the field, future research must prioritize robust, interdisciplinary trials that establish causality and elucidate mechanisms. Overcoming these gaps is essential for developing effective, personalized, and globally relevant nutritional strategies against frailty.

## Figures and Tables

**Figure 1 nutrients-17-03541-f001:**
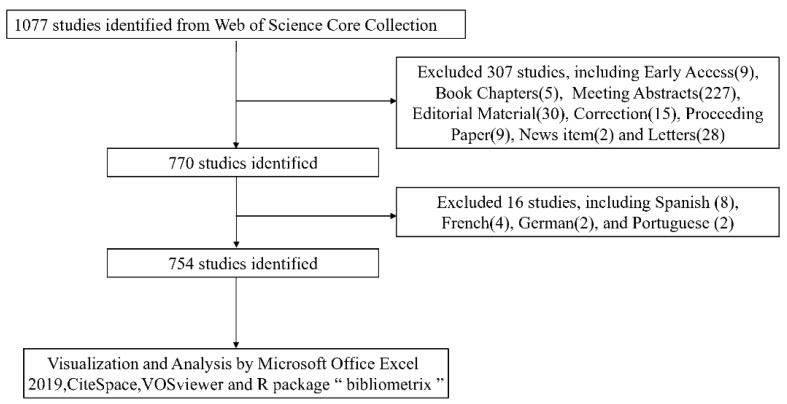
The search and selection workflow.

**Figure 2 nutrients-17-03541-f002:**
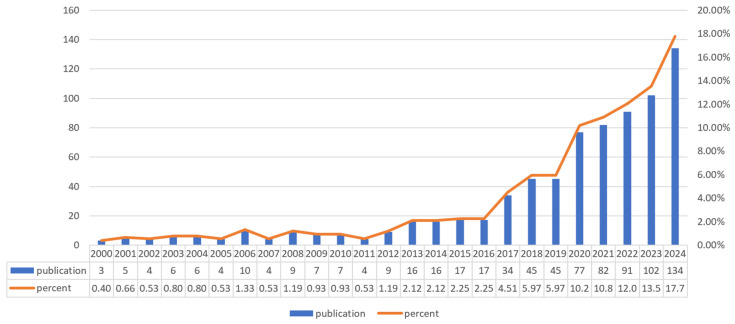
Number of publications in each period. Annual publication counts (blue bars) and their percentage of total publications (orange line) from 2000 to 2024. The left *y*-axis shows publication numbers, and the right *y*-axis shows the percentage of total publications.

**Figure 3 nutrients-17-03541-f003:**
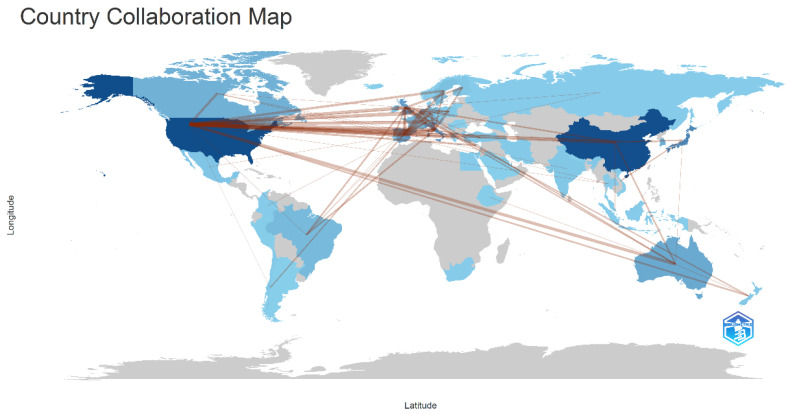
Country collaboration map. Note: The blue nodes represent countries, with the color depth proportional to the publication count of each country. The brown lines indicate collaborative relationships between countries, with thicker lines representing stronger or more frequent cooperation.

**Figure 4 nutrients-17-03541-f004:**
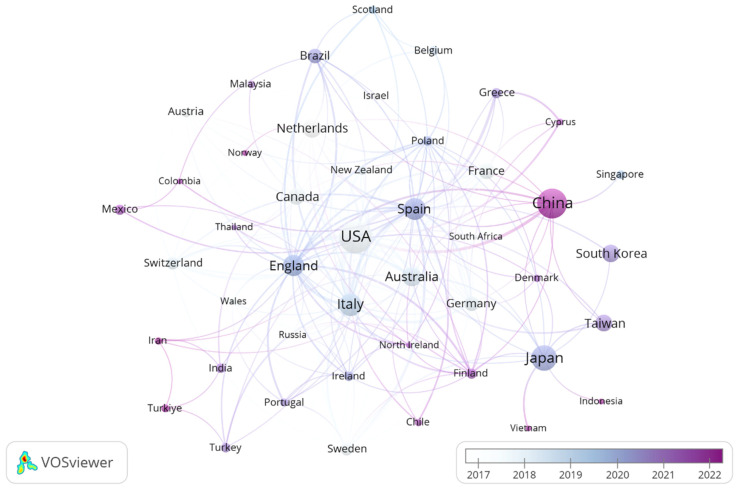
Collaboration network map that changes over time.

**Figure 5 nutrients-17-03541-f005:**
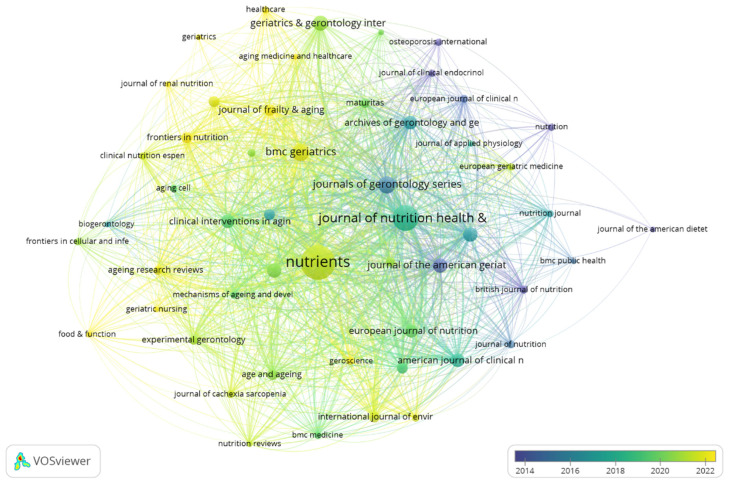
Citation network.

**Figure 6 nutrients-17-03541-f006:**
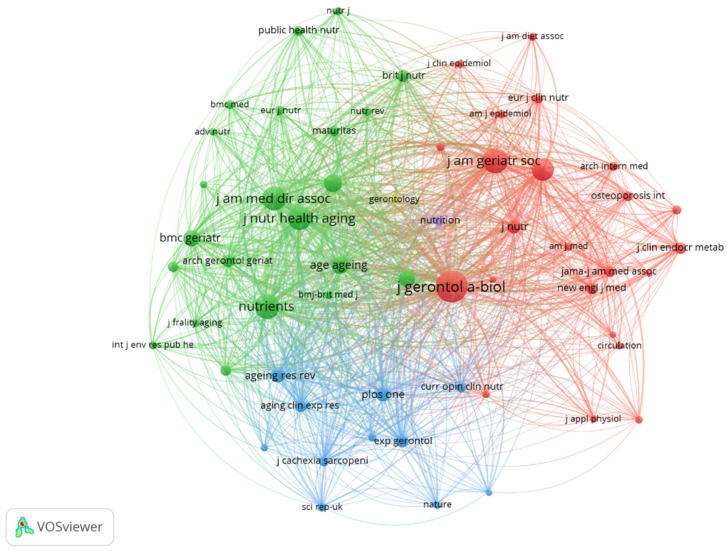
Co-citation network of journals. Note: The colors (green, red, blue) represent distinct clusters formed by the co-citation analysis.

**Figure 7 nutrients-17-03541-f007:**
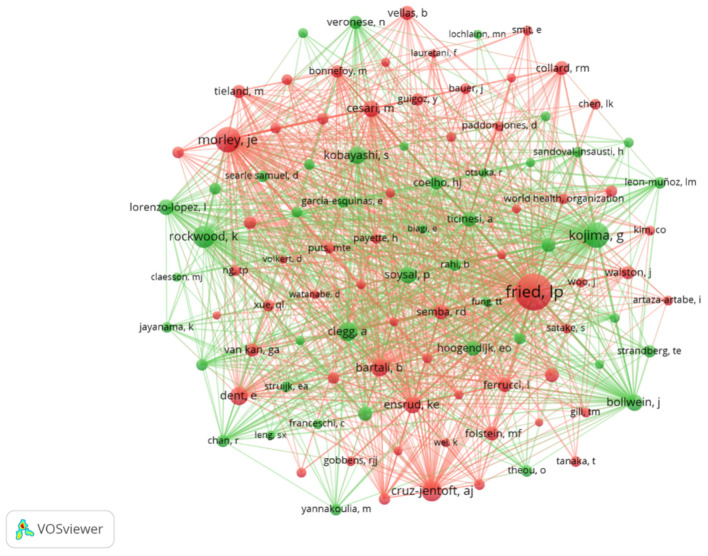
Co-authorship network. Note: Each node represents an author. The size of the node corresponds to the number of publications. The colors represent distinct collaborative clusters of authors, as identified by the VOSviewer clustering algorithm. Links between nodes indicate co-authorship relationships.

**Figure 8 nutrients-17-03541-f008:**
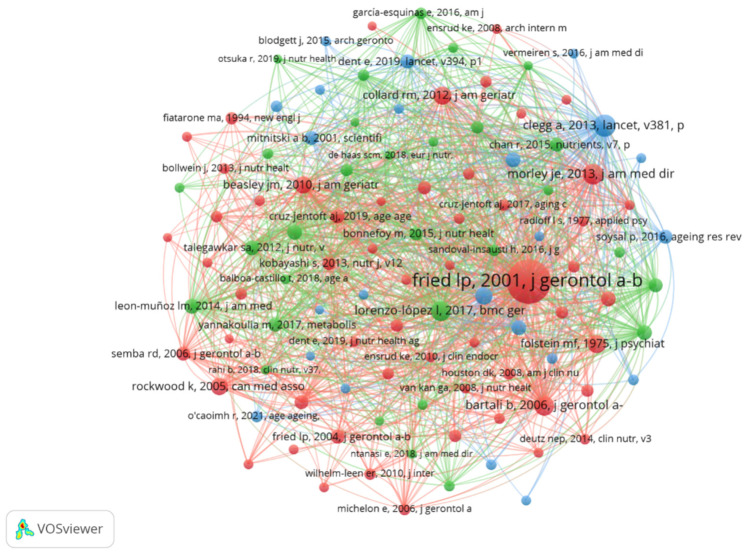
Co-citation network of references. Note: The central and most influential node (highlighted in red/largest size) corresponds to the foundational frailty phenotype paper by Fried et al. [4].

**Figure 9 nutrients-17-03541-f009:**
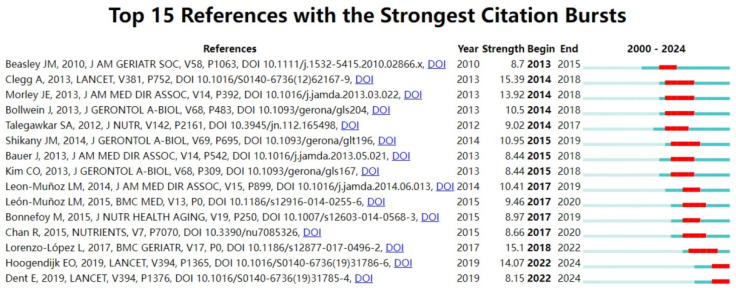
Citation bursts. Note: The blue bars represent the time span of each publication’s presence in the dataset. The red bars indicate the period of a “citation burst,” when the publication was cited with unusually high frequency. The most impactful works include Clegg et al. (2013) [1]. Each bar in the graph represents a year, with red bars indicating strong citation bursts. The earliest citation burst occurred in 2010, while the latest appeared in 2019.

**Figure 10 nutrients-17-03541-f010:**
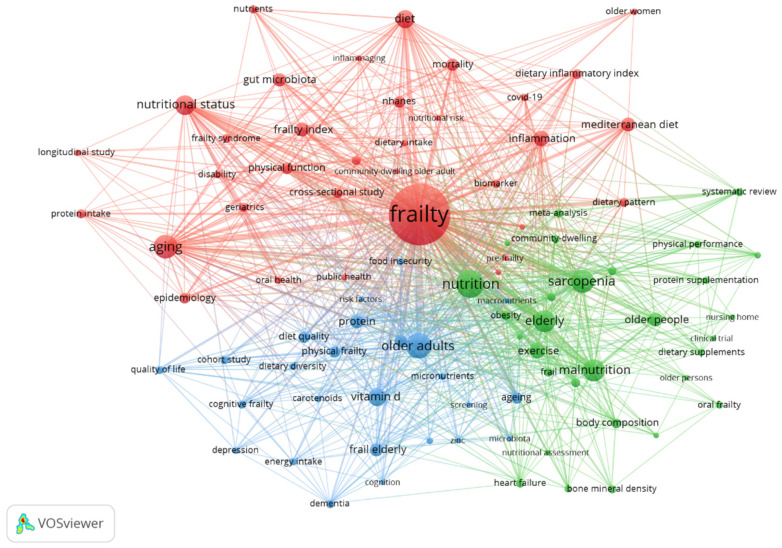
Keyword co-occurrence cluster analysis. Note: The colors represent distinct research clusters identified by the keyword co-occurrence analysis. The red cluster signifies mechanistic and epidemiological investigations, the green cluster represents nutritional interventions and clinical management, and the blue cluster denotes comorbidity studies. The thickness of the lines between nodes indicates the strength of the connections between keywords.

**Figure 11 nutrients-17-03541-f011:**
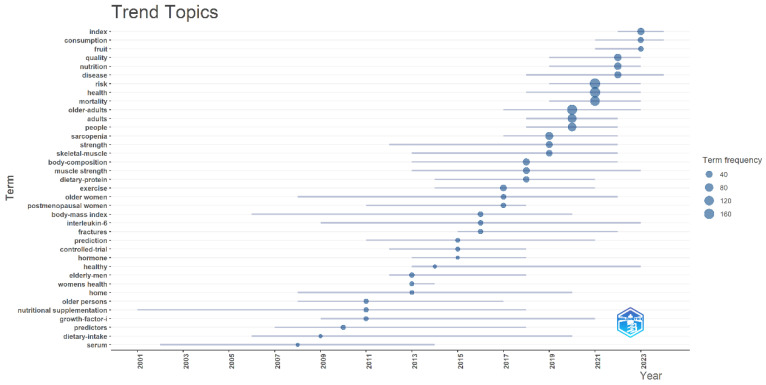
Trend topics.

**Figure 12 nutrients-17-03541-f012:**
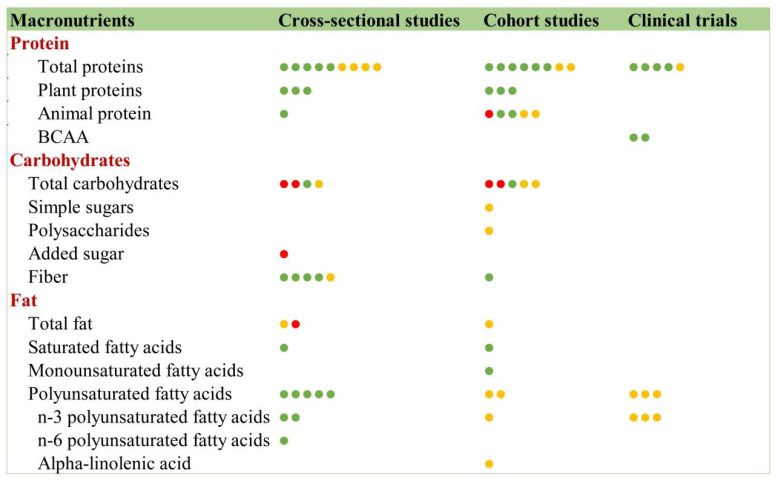
Macronutrients and frailty. Red circles indicate a positive association between the nutrient and frailty, signifying that the nutrient promotes frailty development. Yellow circles denote a neutral association. Green circles represent a negative association, indicating that the nutrient mitigates frailty progression.

**Figure 13 nutrients-17-03541-f013:**
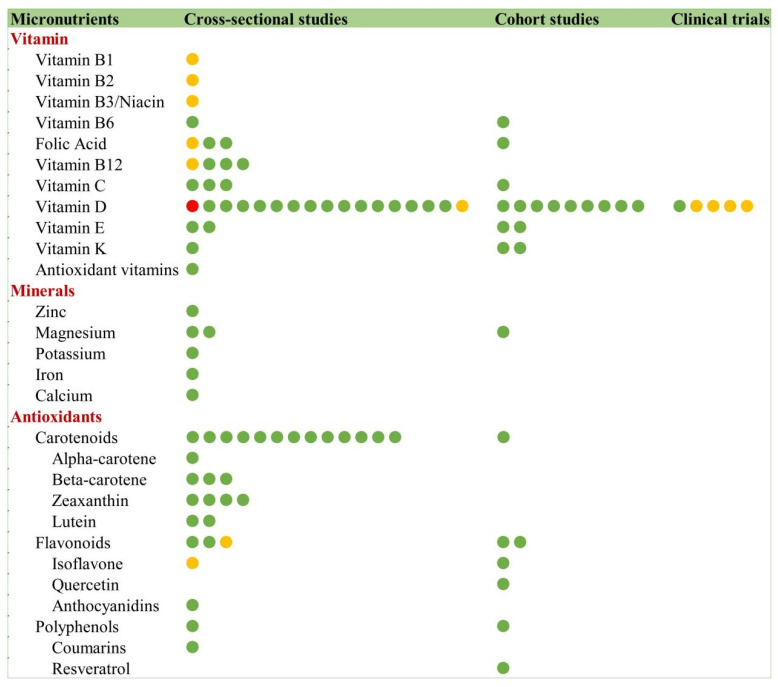
Micronutrients and frailty. Red circles indicate a positive association between the nutrient and frailty, signifying that the nutrient promotes frailty development. Yellow circles denote a neutral association. Green circles represent a negative association, indicating that the nutrient mitigates frailty progression.

**Figure 14 nutrients-17-03541-f014:**
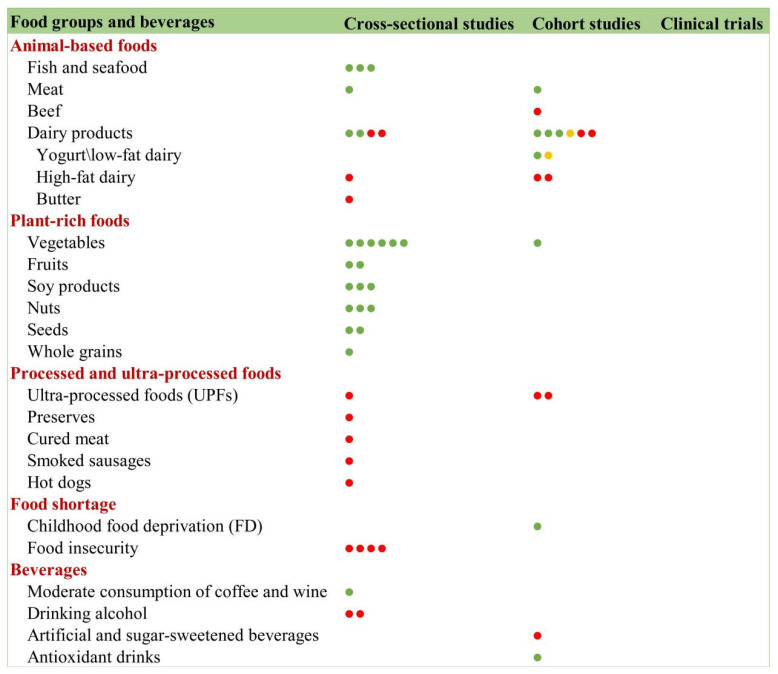
Food groups and frailty. Red circles indicate a positive association between the nutrient and frailty, signifying that the nutrient promotes frailty development. Yellow circles denote a neutral association. Green circles represent a negative association, indicating that the nutrient mitigates frailty progression.

**Figure 15 nutrients-17-03541-f015:**
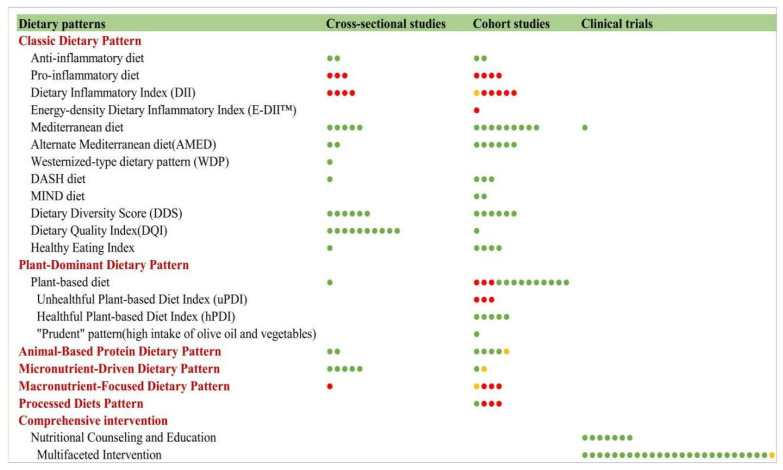
Dietary patterns, comprehensive interventions, and frailty. Red circles indicate a positive association between the nutrient and frailty, signifying that the nutrient promotes frailty development. Yellow circles denote a neutral association. Green circles represent a negative association, indicating that the nutrient mitigates frailty progression.

**Table 1 nutrients-17-03541-t001:** Scientific productivity metrics by nation and institution.

Rank	Country	Counts	Institution	Counts
1	USA (America)	157 (20.8%)	Harvard T.H. Chan School of Public Health	22 (2.9%)
2	China (Asia)	124 (16.4%)	Harvard Medical School	19 (2.5%)
3	Japan (Asia)	87 (11.5%)	National Institute on Aging	15 (1.9%)
4	Italy (Europe)	67 (8.9%)	Universidad Autónoma de Madrid	14 (1.8%)
5	Spain (Europe)	59 (7.8%)	The University of Sydney	14 (1.8%)
6	England (Europe)	54 (7.2%)	Columbia University	13 (1.7%)
7	Australia (Oceania)	44 (5.8%)	Wageningen University and Research	13 (1.7%)
8	Netherlands (Europe)	42 (5.6%)	The Johns Hopkins University	12 (1.6%)
9	South Korea (Asia)	37 (4.9%)	National Health Research Institutes	12 (1.6%)
10	Canada (America)	34 (4.5%)	Ciberesp Ciber Epidemiol and Publ Hlth	11 (1.5%)

**Table 2 nutrients-17-03541-t002:** Top 15 journals by publication count.

Rank	Journal	Counts	IF	Q
1	Nutrients	89 (11.8%)	5.0	Q1
2	The Journal of Nutrition, Health and Aging	52 (6.9%)	4.0	Q1
3	BMC Geriatrics	23 (3.1%)	3.8	Q2
4	Journals of Gerontology Series A: Biological Sciences and Medical Sciences	22 (2.9%)	3.8	Q1
5	Geriatrics and Gerontology International	17 (2.3%)	2.5	Q2
6	Aging Clinical and Experimental Research	16 (2.1%)	3.4	Q2
7	Journal of the American Geriatrics Society	16 (2.1%)	4.3	Q1
8	Journal of the American Medical Directors Association	16 (2.1%)	3.8	Q1
9	Archives of Gerontology and Geriatrics	15 (1.9%)	3.8	Q2
10	The American Journal of Clinical Nutrition	14 (1.8%)	6.5	Q1
11	Clinical Interventions in Aging	14 (1.8%)	3.7	Q3
12	European Journal of Nutrition	13 (1.7%)	4.3	Q1
13	The Journal of Frailty and Aging	11 (1.5%)	3.3	Q2
14	Age and aging	9 (1.2%)	7.1	Q1
15	BMJ Open	9 (1.2%)	2.3	Q2

**Table 3 nutrients-17-03541-t003:** Top 15 co-cited journals.

Rank	Journal	Co-Citation	IF	Q
1	Journals of Gerontology Series A: Biological Sciences and Medical Sciences	2058	4.3	Q1
2	Journal of the American Geriatrics Society	1309	4.5	Q1
3	Nutrients	1216	5.0	Q1
4	The Journal of Nutrition, Health and Aging	1152	4.3	Q1
5	Journal of the American Medical Directors Association	1127	3.8	Q1
6	The American Journal of Clinical Nutrition	1006	6.5	Q1
7	Clinical Nutrition	667	7.4	Q1
8	Age and aging	629	7.1	Q1
9	BMC Geriatrics	556	3.8	Q1
10	The Lancet	525	88.5	Q1
11	PLoS ONE	422	2.6	Q1
12	Aging Research Reviews	414	12.5	Q1
13	The Journal of Nutrition	407	3.7	Q2
14	British Journal of Nutrition	323	3.0	Q2
15	Archives of Gerontology and Geriatrics	313	3.8	Q2

**Table 4 nutrients-17-03541-t004:** Top Ten Authors.

Rank	Authors	Country	Institution	Counts	Co-Cited Authors	Citations
1	Rodriguez–Artalejo, Fernando	Spain	Universidad Autónoma de Madrid	19	Fried, Lp	535
2	Lopez–Garcia, Esther	Spain	Universidad Autónoma de Madrid	14	Morley, Je	248
3	Ferrucci, Luigi	USA	National Institute on Aging (NIA)	10	Kojima, G	245
4	Cesari, Matteo	Italy	University of Milan	9	Rockwood, K	192
5	Veronese, Nicola	Italy	University of Padua	8	Cruz–Jentoft, Aj	166
6	Guallar–Castillon, Pilar	Spain	Universidad Autónoma de Madrid	7	Clegg, A	138
7	Struijk, Ellen A.	Netherlands	Wageningen University	7	Dent, E	125
8	Feart, Catherine	France	University of Bordeaux	7	Kobayashi, S	116
9	Ara, Ignacio	Spain	University of Castilla–La Mancha	6	Bartali, B	113
10	Fung, Teresa T.	USA	Simmons University	6	Cesari, M	109

## Data Availability

The original contributions presented in the study are included in the article/Appendix A; further inquiries can be directed to the corresponding authors.

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
