# Peer review of "Mapping Research Trends in Frailty and Nutrition: A Combined Bibliometric and Structured Review (2000–2024)"

_nutrients, 2025, doi:10.3390/nu17223541_

Round 1

Reviewer 1 Report

Comments and Suggestions for Authors

Review of "Global Evolution of Frailty and Nutrition Research: A Bibliometric Analysis and Hotspot Review (2000-2024)" 

This study investigated that global research trends in nutrition and frailty using a bibliometric analysis and hotspot review. This study has important meaning for future research in these topics. This reviewer only has a few suggestion.

1. In the abstract, please describe more specifically what has been the focus to date and the significant gaps that currently exist.

2. Why wasn't "frailty" included in the search terms?

3. Moreover, "malnutrition" wasn't needed for search terms?

Author Response

1、Comment 1: In the abstract, please describe more specifically what has been the focus to date and the significant gaps that currently exist.

Response 1: We thank the reviewer for this valuable suggestion.  We have revised the abstract to more clearly and specifically outline the current research focus and the identified gaps. The revised parts of the article have been highlighted for your convenience.

2、Comment 2: Why wasn't "frailty" included in the search terms?

Response 2:  We thank the reviewer for raising this important point. In our search strategy, we did include the term “frailty” by using the root “frail” in the title field (TI=(frail)). This approach ensures the retrieval of publications containing “frail” ,“frailty” and related terms in their titles.

3、Comment 3: Moreover, "malnutrition" wasn't needed for search terms?

Response 3: We appreciate the reviewer's insightful suggestion regarding the search term "malnutrition." Our initial search strategy included "malnourished" but not "malnutrition." While "malnourished" captures the adjectival form and is conceptually related, we acknowledge that including "malnutrition" could potentially have retrieved additional relevant records. This is a valuable point for enhancing the comprehensiveness of future searches. In the context of the current study, our search with "malnourished," combined with the broad set of other nutrition-related terms, was designed to capture the core literature at the nutrition-frailty intersection. We have added a note to the limitations section (4.10) to reflect this.

Reviewer 2 Report

Comments and Suggestions for Authors

Dear author,

Thank you to give the opportunity to review your manuscript. I have some suggestions to improve it.

Title

Rephrase the title for precision, avoiding overstated novelty (“A Bibliometric and Structured Review of Frailty and Nutrition Research Trends (2000-2024)”.

Abstract

Temper claims to being “first” or “comprehensive”, clarify limits (single database, English-only). Rephrase casual statement to “associated with” unless supported by RCTs. Specify exactly what was analysed, which tools/software, and database.

Introduction

Ensure all definitions and claims are clearly referenced. Provided a structured paragraph summarizing previous bibliometric analyses to position the study properly.

Methods

Quantify excluded articles by language and discuss potential bias.

Specify inclusion/exclusion criteria transparently, including thresholds, types of studies, and instruments.

State all software settings, including normalization, clustering, and citation burst thresholds.

Results

Move PRISMA results form methods to results.

Provide complete parameter settings for citation burst analysis, clarify and include all figures/tables as actual appendices for reproducibility.

Clearly mark study design for each finding (RCT, cohort, cross-sectional). Use “associated with” for observational associations. Indicate whether recommendations are hypothesis-generating or evidence-based.

Present limitations of small/contradictory RCTs and cohort studies. Highlight the need for more robust interventional evidence. Avoid blanket recommendations not supported by systematic review/meta-analysis.

Specify study design for each claim. Flag areas with insufficient evidence and avoid overgeneralizing based on cross-sectional results. Suggest further research where gaps exist.

Discussion

Explicitly discuss the limitations of observational predominance and lack of robust RCTs for many nutrients.

Eliminated novelty claims not substantiated by systematic background search.

Standardize the tone to match academic rigor.

Limitations

Reflect on the potential impact on global representativeness and synthesize how these may limit conclusions.

Conclusions

Soften conclusions to highlight the field’s rapid growth, predominance of associative evidence, gaps in interventions and global collaboration. Point to the need for robust protocols and greater interdisciplinary integration, as well as reproducible and accessible future research.

References

Standardize all citations according to journal guidelines, correct inconsistencies, and verify DOIs, author names, and pagination.

Figures/Tables

Ensure all are present, properly numbered, and readable. Supply supplementary datasets and scripts for all analyses.

Best regards,

Comments on the Quality of English Language
  • There are occasional grammatical and syntactical errors such as missing articles, awkward constructions, verb tense inconsistencies, and prepositional misuse that reduce fluency.

  • Some sentences are overly long or complex, which impacts readability; breaking them into shorter, clearer statements would help.

  • Terminology and abbreviations are sometimes inconsistently used and should be standardized.

  • The manuscript would benefit from a thorough English language editing pass focusing on grammar, syntax, clarity, concision, and consistency.

  • Tense should be harmonized across sections (usually past tense for methods/results, present for accepted knowledge).

  • Accessibility could improve with clearer definitions or simpler phrasing of complicated scientific concepts.

Author Response

1、Comment 1: Tiltle: Rephrase the title for precision, avoiding overstated novelty (“A Bibliometric and Structured Review of Frailty and Nutrition Research Trends (2000-2024)”).

Response 1: We sincerely thank the reviewer for this insightful suggestion. We agree that the original title could be more precise. In response, we have rephrased the title to "Mapping Research Trends in Frailty and Nutrition: A Combined Bibliometric and Structured Review (2000-2024)". This new title removes the overstated "Global Evolution" and "Hotspot Review" as advised, and instead precisely highlights the two core methodologies of our study—bibliometric analysis and structured review—by incorporating the term "Structured Review" and framing the bibliometric aspect with the well-established term "Mapping Research Trends". We believe this revision accurately and modestly reflects the content of our work.

2、Comment 2: Abstract: Temper claims to being “first” or “comprehensive”, clarify limits (single database, English-only). Rephrase casual statement to “associated with” unless supported by RCTs. Specify exactly what was analysed, which tools/software, and database. To facilitate the review process, we have kindly highlighted all modifications in the revised manuscript.

Response 2: We sincerely thank the reviewer for these crucial suggestions to enhance the precision and rigor of our abstract. We have thoroughly revised the abstract accordingly:

  • Tempered Claims: We have removed the terms "first," "comprehensive," and "inaugural" throughout the abstract (e.g., in Background and Conclusion) to present our findings in a more measured manner.

  • Clarify limits: The limits of our study have been clarified by explicitly stating in the Limitations that only publications from the Web of Science Core Collection (WoSCC) were analyzed, with English language as the only inclusion criterion.

  • Rephrase casual statement: We fully agree that in the absence of evidence from randomized controlled trials (RCTs), the language should accurately reflect associations rather than causality. As suggested, we have carefully reviewed the Abstract and rephrased the relevant statements to ensure scientific precision.
  • Specify exactly what was analysed, which tools/software, and database: As recommended, we have revised the "Methods" section in the Abstract to provide more specific details. 

3、Comment 3: Introduction: Ensure all definitions and claims are clearly referenced. Provided a structured paragraph summarizing previous bibliometric analyses to position the study properly.

Response 3: We sincerely thank the reviewer for this constructive suggestion. We have thoroughly revised the Introduction section to address both points raised.

  • Referencing of Definitions and Claims: We have carefully reviewed the entire Introduction to ensure that all key definitions (e.g., the concept of frailty, its prevalence, and its association with nutrition) and substantive claims are now supported by appropriate citations. We have added references where necessary to enhance the academic rigor and credibility of our background presentation.
  • Provided a structured paragraph: We sincerely thank the reviewer for this valuable suggestion. As recommended, we have added a new paragraph in the Introduction section (highlighted in yellow) that systematically reviews previous bibliometric studies in adjacent fields (such as cognitive aging and sarcopenia). This addition clearly delineates the existing research landscape and explicitly identifies the knowledge gap that our study aims to fill—namely, the absence of a dedicated bibliometric analysis at the critical intersection of frailty and nutrition. We believe this revision significantly strengthens the positioning and justification for our study.

4、Comments 4: Methods: Quantify excluded articles by language and discuss potential bias. Specify inclusion/exclusion criteria transparently, including thresholds, types of studies, and instruments. State all software settings, including normalization, clustering, and citation burst thresholds.

Response 4: We sincerely thank the reviewer for these insightful and constructive suggestions, which have significantly improved the methodological rigor and transparency of our study. We have thoroughly revised the Methods section accordingly:

  • Language Exclusion and Bias: As suggested, we have now quantified the number of non-English publications excluded (n=16) and added a discussion about the potential bias this may introduce in the Limitations section (Section 4.10).

  • Inclusion/Exclusion Criteria: We have substantially expanded Section 2.2 to provide a fully transparent account of our criteria. This includes specifying the hierarchical preference for study designs, listing all frailty assessment instruments considered, and detailing the rationales for exclusion (e.g., specific terminal illnesses, combined interventions).

  • Software Settings: We have created a new subsection, 2.3.1 Bibliometric Software and Parameters, which explicitly states all critical settings and thresholds used in VOSviewer, CiteSpace, and the Bibliometrix R package, including the normalization method, clustering resolution, and citation burst detection parameters.

5、Comments 5: Results:

Move PRISMA results form methods to results.

Provide complete parameter settings for citation burst analysis, clarify and include all figures/tables as actual appendices for reproducibility. Clearly mark study design for each finding (RCT, cohort, cross-sectional).

Use “associated with” for observational associations. Indicate whether recommendations are hypothesis-generating or evidence-based.

Present limitations of small/contradictory RCTs and cohort studies. Highlight the need for more robust interventional evidence. Avoid blanket recommendations not supported by systematic review/meta-analysis.

Specify study design for each claim. Flag areas with insufficient evidence and avoid overgeneralizing based on cross-sectional results. Suggest further research where gaps exist.

Response 5: We have carefully considered all the points raised and have revised the manuscript accordingly. We believe that the revisions have significantly strengthened the clarity, rigor, and overall quality of our work.

  • We agree that presenting the literature search and selection flow (PRISMA results) in the Results section improves the manuscript's structure and clarity. As suggested, we have now moved the detailed description of the search strategy, inclusion/exclusion criteria, and the PRISMA flowchart from the Methods section (previously subsections 2.1 and 2.2) to the Results section, where it now forms the new subsection 3.1 ("Literature Search and Selection"). The corresponding figures and text have been updated accordingly.
  • The complete parameter settings for the citation burst analysis are now detailed in the Methods section (subsection 2.3.1). All supplementary figures and tables (e.g., Appendix Figure 1, Appendix Table 1, etc.) are now provided in the Appendix section following the References. A statement confirming this has been added to the Methods (2.3.1).
  • We have carefully revised the manuscript according to your suggestions. Specifically, in Section 3.9 (Key Findings of Nutrition and Frailty), we have now: Clearly indicated the study design (e.g., RCT, cohort, cross-sectional) for each finding; Used “associated with” to describe observational associations; Distinguished between hypothesis-generating observations and evidence-based recommendations.
  • We thank the reviewer for this critical suggestion. In response, we have substantially revised the Discussion (Section 4.8) to explicitly present the evidence limitations. We now synthesize these around three core asymmetries: a design imbalance (dominance of observational data over scarce RCTs), an intervention efficacy gap (e.g., vitamin D's associative vs. interventional evidence), and a fragmentation of evidence for minerals and food groups. This framework directly highlights the limitations of the current evidence and underscores the need for more robust interventional studies. We have consequently avoided any blanket recommendations, ensuring all conclusions are strictly aligned with the evidence from our review.
  • We have fully incorporated this guidance. The study design (e.g., RCT, cohort) is now specified for each finding in Section 3.9. In the Discussion (Section 4.8), we explicitly flag areas with insufficient evidence (e.g., minerals, specific food groups) using cautious language such as "hypothesis-generating" and "fragmentary" to avoid overgeneralization. These identified gaps directly inform our specific proposals for future research, which prioritize mechanistic RCTs and standardized methodologies to address these shortcomings.

6、Comments 6: Discussion

Explicitly discuss the limitations of observational predominance and lack of robust RCTs for many nutrients.

Eliminated novelty claims not substantiated by systematic background search.

Standardize the tone to match academic rigor.

Response 6: We thank the reviewer for these insightful comments. We have revised the manuscript accordingly:

  • Observational vs. RCT Evidence: We have explicitly discussed the overreliance on observational data and the critical lack of robust RCTs for many nutrients, particularly in a new paragraph in Section 4.8 (Key Findings). This includes quantifying the prevalence of study designs and highlighting the efficacy-evidence gap using Vitamin D as a key example.

  • Novelty Claims: All unsubstantiated claims of novelty have been removed from the manuscript, particularly in the Introduction and Strengths sections. The study's contribution is now framed more accurately within the existing literature.

  • Academic Tone: The tone has been standardized throughout the manuscript to ensure it meets the expected level of academic rigor, especially in the Discussion and Conclusion sections.

7、Comments 7: Limitations

Reflect on the potential impact on global representativeness and synthesize how these may limit conclusions.

Response 7: We sincerely thank the reviewer for this valuable suggestion. We have thoroughly revised the ‘Limitations’ section (now Section 4.10) to directly address this point. Specifically, we have added a detailed reflection on how the reliance on a single database (Web of Science) and the restriction to English-language publications may introduce database and language biases, potentially skewing our findings toward Anglophone and high-income country research paradigms. We further synthesize that these limitations caution against overgeneralizing our conclusions and explicitly state that the identified research gaps (e.g., lack of global diversity) are themselves influenced by these methodological constraints. The revised text concludes by suggesting future directions to overcome these limitations. We believe these additions significantly strengthen the manuscript by providing a more critical and nuanced interpretation of our findings.

8、Comments 8: Conclusions

Soften conclusions to highlight the field’s rapid growth, predominance of associative evidence, gaps in interventions and global collaboration. Point to the need for robust protocols and greater interdisciplinary integration, as well as reproducible and accessible future research.

Response 8: We sincerely thank the reviewer for this constructive suggestion. We have revised the Conclusions to better highlight the field's rapid growth, the current predominance of associative evidence, and the critical gaps in interventional studies and global collaboration. The revised conclusion now more clearly points to the need for robust, interdisciplinary trials to develop effective and globally relevant strategies. We believe these edits have strengthened the manuscript.

9、Comments 9: References

Standardize all citations according to journal guidelines, correct inconsistencies, and verify DOIs, author names, and pagination.

Response 9: Thank you for your valuable feedback. We have thoroughly revised the reference list to conform to standard formatting guidelines as requested. We appreciate your guidance in improving our manuscript.

10、Comments 10: Figures/Tables

Ensure all are present, properly numbered, and readable. Supply supplementary datasets and scripts for all analyses.

Response 10: Thank you for your valuable suggestion. We have added Data Availability Statement to the manuscript to enhance transparency and reproducibility. The statement clarifies that all data supporting the findings are included within the article and supplementary materials, with further inquiries directed to the corresponding author. All the tables and charts in the article have been properly marked.

11、Comments 11: Grammatical and syntactical errors; some sentences are overly long or complex; Terminology and abbreviations are sometimes inconsistently used; Tense should be harmonized across sections,etc.

Response 11: We have carefully revised the manuscript to enhance its overall quality, with particular attention to the grammatical accuracy, sentence conciseness, consistency of terms, unity of tenses and overall clarity.

Reviewer 3 Report

Comments and Suggestions for Authors

The general idea of the study is undoubtedly both current and interesting. Yet, methodological flaws (even if small in number) seriously undermine the credibility of the analysis presented.

  • Section 2.2.1 states that cross-sectional studies were included in the analysis (and that with priority). Furthermore, inclusion criteria comprise “clearly defined exposure factors or nutritional interventions with appropriate control groups”
  • Section 2.2.2 lists under exclusion criteria: “studies lacking control groups,” “insufficient sample sizes,” or “inadequate statistical adjustment for confounders.”

The authors, first off, must clarify a few things:

  • What kind of cross-sectional studies were found following nutritional interventions, with appropriate control groups? This is a contradiction.
  • “Insufficient sample size” must be formally defined for the retrieval procedure.
  • The same applies to “inadequate statistical adjustment for confounders,” which is not defined.

I am very much interested in reviewing an amended version of the manuscript. Yet, to be able to relate to the analysis substantively, the above points must be addressed.

A minor point: in the sentence “The search and selection workflow are illustrated in the figure,” Figure 1 is actually “above.”

Author Response

1、Comment 1: Clarification on the inclusion of cross-sectional studies and the requirement for control groups. Response 1: We sincerely thank the reviewer for highlighting this ambiguity. The reviewer is correct that the requirement for a "control group" is specific to interventional studies and does not directly apply to observational designs like cross-sectional studies. To resolve this contradiction and enhance methodological transparency, we have revised the Inclusion Criteria (Section 2.2.1). We have now clearly separated the criteria for observational and interventional studies. Specifically, we now require "clearly defined nutritional exposures" for observational studies and "nutritional interventions with an appropriate control group" for interventional studies. This clarification ensures logical consistency in our study selection process.

Changes in the manuscript: In Section 2.2.1, criterion (3) has been split and rephrased as follows: “(3) Exposure/Intervention: for observational studies, a clearly defined nutritional exposure (e.g., intake levels of specific nutrients, food groups, or dietary patterns) was required. For interventional studies, a clearly defined nutritional intervention (e.g., supplementation) with an appropriate control or comparison group was required.”

2、Comment 2: Formal definition of "insufficient sample size."
Response 2: We agree with the reviewer that a formal definition was needed to ensure objectivity and reproducibility. Following the reviewer's suggestion, we have now defined explicit, quantitative thresholds for sample size exclusion in the Exclusion Criteria (Section 2.2.2). These thresholds are based on common benchmarks used in similar nutritional epidemiology and geriatric research. 

3、Comment 3: Formal definition of "inadequate statistical adjustment for confounders."
Response 3: This is an excellent point. To eliminate subjectivity, we have specified the core set of confounders that we considered essential for adequate adjustment in studies of nutrition and frailty. This provides a clear and justifiable basis for our assessment.
Changes in the manuscript: In Section 2.2.2, criterion (1) has been further clarified to read: ”inadequate statistical adjustment for key confounders (as a minimum, age and sex; ideally also including total energy intake, comorbidities, and socioeconomic status).”

4、Comment 4: Minor point regarding the reference to Figure 1.
Response 4: We thank the reviewer for their attention to detail. The text has been corrected for accuracy.

Round 2

Reviewer 1 Report

Comments and Suggestions for Authors

No further comments.

Reviewer 2 Report

Comments and Suggestions for Authors

Dear authors,

The changes have been successfully implemented.

Kind regards,